# Technology Transfer of O-(2-[18F] Fluoroethyl)-L-Tyrosine (IASOglio^®^) Radiopharmaceutical

**DOI:** 10.3390/ph18060769

**Published:** 2025-05-22

**Authors:** Anna Notaro, Salvatore Limpido, Lucie Plougastel, Alessandro Zega, Mauro Telleschi, Mauro Quaglierini, Alessia Danti, Antonio Fiore, Letizia Guiducci, Michela Poli

**Affiliations:** 1CURIUM SPECT Europe (Cis Bio International), CMC Department of R&D, 91400 Saclay, France; anna.notaro@curiumpharma.com (A.N.); lucie.plougastel@curiumpharma.com (L.P.); 2Curium Italy, 20155 Milano, Italy; salvatore.limpido@curiumpharma.com (S.L.); alessia.danti@curiumpharma.com (A.D.); 3Institute of Clinical Physiology, National Research Council, 56124 Pisa, Italy; alessandro.zega@cnr.it (A.Z.); mauro.telleschi@cnr.it (M.T.); mauro.quaglierini@cnr.it (M.Q.); antoniofiore@cnr.it (A.F.); letizia.guiducci@cnr.it (L.G.)

**Keywords:** radiopharmaceuticals, glioma, [^18^F]FET, GMP, technology transfer

## Abstract

**Background/Objectives**: Gliomas, including the most aggressive subtype—glioblastoma multiforme, are brain tumors with an unfavorable prognosis and high mortality. Early diagnosis is essential to improve treatment efficacy. Positron emission tomography PET with O-(2-[^18^F] fluoroethyl)-L-tyrosine ([^18^F]FET) has been supported by clinical studies for its role in diagnosis and monitoring the disease. However, the low availability of [^18^F]FET in Italy has limited its use in clinical praxis. This study describes the technological transfer of the radiopharmaceutical IASOglio^®^ (the commercial [^18^F]FET developed by Curium Pharma in Italy), with the aim of improving national access to this advanced diagnostic technology. **Methods**: Three consecutive batches were produced using the automated Trasis AllinOne module, and quality control was performed, including chemical and microbiological tests, to successfully validate the production process. Additionally, the stability of the radiopharmaceutical for its entire shelf life has been demonstrated with stability testing at 14 h after end of synthesis (EOS). **Results**: The production of [^18^F]FET achieved a non-corrected yield between 49% and 52%, with a corrected decay rate ranging from 73% to 79%. The process met the required quality specifications, including bio-burden control and filter integrity. The technological transfer was successfully completed, and production authorization was obtained from the Italian Medicines Agency (AIFA) for the Officina Farmaceutica of Institute of Clinical Physiology of the National Research Council (CNR-IFC) located in Pisa. **Conclusions**: Local production of [^18^F]FET in Italy marks a milestone in glioma diagnosis, thereby contributing to timely treatment and improved clinical outcomes.

## 1. Introduction

Gliomas, ranging from low-grade to high-grade—including the rapidly growing and aggressive subtype glioblastoma multiforme (GBM)—represent the most common malignant primary brain tumors, with an incidence of 4–5/100,000 individuals per year. Gliomas are the second cause of cancer mortality in adults under the age of 35 and the fourth cause in subjects younger than at 54 years old, claiming the lives of approximately 13,770 people per year in the United States. Glioblastoma (GBM) is the aggressive, fast-growing, high-grade form and accounts for more than half of gliomas with a median survival rate of approximately 15 months and a 5-year survival rate of approximately 5% [1,2].

The diagnostic potential of O-(2-[^18^F] fluoroethyl)-L-tyrosine ([^18^F]FET) positron emission tomography (PET) has already been proven in many studies and, in addition to the golden standard magnetic resonance imaging (MRI), provides important additional information for diagnosis, grading, follow-up, and choice of therapy for patients suffering from brain tumors [3,4,5].

The use of amino acid PET as an imaging modality for gliomas, complementing MRI, is encouraged by the European Association of Nuclear Medicine (EANM), the Society of Nuclear Medicine and Molecular Imaging (SNMMI), the European Association of Neurooncology (EANO), and the working group for Response Assessment in Neurooncology with PET (PET-RANO), which have drawn up two guidelines for imaging of gliomas using PET with radio-labeled amino acids and 2-Deoxy-2-[^18^F]fluoroglucose ([^18^F]FDG) [1,6].

As reported by Albert et al. [1], compared to MRI alone, [^18^F]FET has demonstrated superior diagnostic accuracy in differentiating glioma from nonneoplastic lesions, grading glioma, and differentiating glioma recurrence from treatment-induced alterations (e.g., pseudoprogression, radionecrosis). In addition, [^18^F]FET, exhibited higher diagnostic accuracy than MRI in newly diagnosed glioblastoma and metabolically active WHO grade II/IV gliomas. Ultimately, [^18^F]FET is superior to MRI in the assessment of treatment response and of prognosis in gliomas.

A limitation in the routine use of [^18^F]FET is the very low availability of the radiotracer. In Italy, commercial [^18^F]FET is not available, and its clinical use is restricted to a few nuclear medicine centers with a cyclotron and radiopharmacy or to imported radiopharmaceuticals from abroad.

It is worth noting that the adoption of production processes compliant with Good Manufacturing Practice (GMP) standards is essential to guarantee the quality of batches for large-scale distribution [7], and the short half-life of radionuclides (with all its constraints) makes the process validation of a PET radiopharmaceutical very complex.

At present in Europe, the radiopharmaceutical [^18^F]FET, produced according to GMP standards, is commercialized as IASOglio^®^, via a Marketing Authorization (MA) owned by Curium PET France [8] (see Appendix A). [^18^F]FET is one of the first 18F-labeled amino acids for the study of metabolism in tumors, and, although the first synthesis procedure was developed several years ago, its synthetic methods still present difficulties, especially due to the need for a two-step reaction, low yields, and long synthesis times (>60 min) [9].

Hamacher and Coenen [10] and Wester et al. [11] proposed two types of synthesis of [^18^F]FET using different precursors: Hamacher and Coenen employed direct nucleophilic radiofluorination of the protected precursor, while Wester used a two-step reaction. Both methods have been optimized for implementation in remote-controlled synthesizers that meet GMP conditions [12].

More in detail, radiosynthesis of [^18^F]FET can be achieved by nucleophilic fluorination of ethylene glycol-1,2-ditosylate and subsequent fluoroethylation of the unprotected precursor L-tyrosine disodium salt. However, the radiochemical yield of this method is low due to the double purification. This synthesis has a duration of 60 min and a 40% yield [11].

An alternative method uses TBA for the elution of [^18^F]-fluoride in the reactor, and it is based on direct nucleophilic fluorination of the protected precursor O-(2-tosyloxyethyl)-N-trityl-L-tyrosine tert-butyl ester (TET) followed by hydrolysis with trifluoroacetic acid (TFA). This method has shown higher synthesis yields but has the disadvantage of using the corrosive and toxic 1,2-dichloroethane (TFA), as well as a long synthesis time (about 80 min) with a 60% yield [10].

The synthesis method adopted in the [^18^F]FET (IASOglio^®^) Marketing Authorization provides direct radiolabeling of the precursor TET and subsequent hydrolysis in acidic conditions.

The synthesis was developed on the Trasis AllinOne module from an existing synthesis in the Trasis synthesis portfolio, with modifications to the purification step and formulation of the final product. The conditions of the synthesis steps already optimized by Trasis were maintained [13]. It is worth emphasizing that optimization of the synthesis steps is crucial for the commercial production of radiopharmaceuticals with a medium-to-short lifetime to ensure that sufficient active compound is available for clinical use. Indeed, a well-optimized process minimizes synthesis times while maintaining the efficiency and effectiveness of the synthesis process, allowing the radiopharmaceutical to be exploited for longer periods of time.

The Chemistry Manufacturing Control (CMC) division of the R&D team of Curium focused on the optimization of purification, identifying the best HPLC column for isolating [^18^F]FET, as well as the preferred injection conditions on the column (pH, composition of the injected solution, and volume), to obtain a product complying with the specifications reported in the European Pharmacopeia. A second part of the work focused on the formulation of the finished product, with the aim of obtaining a product stable throughout its shelf life at the activity concentration described in the MA dossier [14].

Following the MA update, the CMC division of the R&D team of Curium organized the deployment of the [^18^F]FET (IASOglio^®^) production process and quality control methods to European manufacturing sites through the technology transfer process, which is the first step towards the commercial production of a radiopharmaceutical in a PET production site.

In Italy, the first production site that carried out the technology transfer of [^18^F]FET (IASOglio^®^) was the Officina Farmaceutica of Institute of Clinical Physiology of the National Research Council (CNR-IFC) Pisa PET production site. The production site was authorized by the Italian Medicines Agency (Agenzia Italiana del Farmaco, AIFA), and the MA dossier variation is currently on-going.

In this paper we describe the technology transfer process of [^18^F]FET (IASOglio^®^) in Italy, within a collaboration between the Italian Research Public Institution (CNR) and Curium, born with the aim of making a wider choice of radiopharmaceuticals available to the community.

## 2. Results

For [^18^F]FET (IASOglio^®^) process validation, three consecutive batches were manufactured at the CNR-IFC Pisa PET production site under standard laboratory conditions, with a batch size from 28.6 mL to 57.5 mL and total product activity from 56.7 to 113.7 GBq at calibration time.

The three validation batches had a yield, without decay correction, between 49 and 52% and a radio decay-corrected yield between 73 and 79%. The decay-corrected yield provides standardized information about the synthesis regardless of the decay time of the radioisotope. The decay-corrected yield is calculated using the following formula:A_t_ = A_0_ × e^(−λt)^.
where

A_t_ = corrected activity;

A_0_ = initial activity;

λ = decay constant (ln2/Half-life(s)) for ^18^F λ = 1.05 × 10^−4^ (s^−1^);

t = time elapsed (s).

The most critical physical and chemical properties of the product were measured to evaluate the conformity of the final radiopharmaceutical, and some of them were monitored at expiry to ensure the stability of the commercial product as described in the MA.

Chemical and radiochemical purity are critical parameters for ensuring the quality of the imaging. Radiochemical purity (RCP) was systematically over 95%, within a linearity range previously defined based on the highest and lowest calibration described in the MA dossier. Chemical purity analysis of the [^18^F]FET (IASOglio^®^) manufacturing process systematically resulted in [^19^F]-FET content lower than 10 µg/mL, as requested by the European Pharmacopeia monograph.

Residual solvents, excipients, and possible chemical and long-lasting impurities were also verified in accordance with the MA dossier to guarantee a controlled amount of eventual toxic agents within the safety limits. The analytical methods and the used instrumentations are reported in the “Materials and Methods” Section.

In addition, the sterility of the process and the absence of microbiological growth were verified, ensuring the safety of the final solution for patient injection.

The results of the three validation batches, analyzed in accordance with the European Pharmacopeia and IASOglio dossier, are reported in Table 1.

Ultimately, to ensure the stability of the IASOglio^®^ drug product during the entire shelf life (14 h), a new set of tests were performed according to the validation stability protocol (Table 2).

Three batches of the IASOglio^®^ drug product were manufactured for the bioburden evaluation. The results obtained for the process are summarized in Table 3.

In conclusion, all the results obtained regarding the in-process controls and the final product analyses at release and expiry confirmed that the [^18^F]FET (IASOglio^®^) manufacturing process reproducibly leads to a product that conforms to specifications (MA dossier) and is aligned with, or equivalent to, the European Pharmacopeia throughout its entire shelf life, on the CNR-IFC Pisa PET production site.

## 3. Discussion

Gliomas, though they represent only 3% of all malignant brain tumors, are among the most aggressive and unfavorable, contributing to the greatest number of years of life lost compared to other neoplasms. The impact on the patient’s life and the broader socio-economic sphere is devastating. Early diagnosis remains the only opportunity for successful intervention.

The role of [^18^F]FET-PET in the diagnosis, follow-up, and management of gliomas has been well established in clinical studies, underscoring its value in early detection, diagnosis of recurrence, monitoring of treatment, and appropriate therapy selection. Its application is widely supported by expert guidelines, reinforcing its importance in clinical practice. However, access to [^18^F]FET has historically been limited in Italy, primarily due to its restricted availability and the necessity for importation, making it difficult for many patients to benefit from this advanced diagnostic tool.

The availability of [^18^F]FET at a national level offers a new era of precision medicine for glioma patients in Italy. Every year, over 6500 individuals are diagnosed with gliomas, and with the expanded access to [^18^F]FET, more patients will benefit from early and accurate diagnoses. This will ultimately lead to improved clinical outcomes through better-informed treatment decisions and more personalized care. Early detection is key to increasing the effectiveness of treatments, improving survival rates, and enhancing the overall quality of life for patients affected by gliomas.

Furthermore, the broader availability of [^18^F]FET for glioma diagnosis will help mitigate the socio-economic impact of these malignancies by enabling earlier intervention. By facilitating timely diagnosis and treatment, the healthcare system will be better equipped to address the rising prevalence of gliomas, reducing the long-term costs associated with delayed diagnoses and wrong or advanced-stage treatments. This will contribute to a more sustainable healthcare system, where economic burdens are minimized.

In addition to the direct benefits to patients, this expansion of [^18^F]FET availability highlights Italy’s commitment to enhancing its cancer care infrastructure. The integration of cutting-edge diagnostic technologies, such as [^18^F]FET-PET, represents a significant step forward in aligning with global best practices for the diagnosis and management of patients with gliomas. It underscores the importance of continuous innovation in healthcare, ensuring that patients in Italy receive the best possible care, irrespective of geographic location.

The optimization of the manufacturing process has reduced the synthesis time to 60 min significantly improving the availability of the radiotracer (half-life of fluorine-18 is equal to 109.771 min) while maintaining the efficiency and effectiveness of the synthesis process.

The CNR-IFC Pisa PET production site was the first Italian site to successfully complete the technology transfer of [^18^F]FET (IASOglio^®^), providing documented evidence that the process reliably and reproducibly delivers a final product that conforms to specifications, adhering to the process and maintaining process parameters within the examined range (Table 1 and Table 2).

The successful completion of the technology transfer clears the way for the regulatory process that will ultimately lead to the commercial production of [^18^F]FET (IASOglio^®^) by the CNR-IFC Pisa PET production site.

First, the AIFA issued a GMP certificate to the CNR-IFC Pisa PET production site, which is the first Italian site to receive authorization to produce [^18^F]FET (IASOglio^®^), in accordance with the indication provided by Curium Pharma. Then, the request to add the CNR-IFC Pisa PET production site to the current Marketing Authorization was submitted and approved in January 2025. A Mutual Recognition Procedure (MRP), whereby a marketing authorization granted in one member state can be recognized in other EU countries, is currently underway to allow commercial production of [^18^F]FET (IASOglio^®^).

Once the regulatory process is completed, the CNR-IFC Pisa PET production site will be authorized to distribute [^18^F]FET (IASOglio^®^), ensuring access to medical treatment for all citizens.

In conclusion, this technology transfer provides an important contribution to improving the clinical accuracy of imaging tools for the diagnosis and follow-up of oncological patients and constitutes an additional opportunity for accurate and personalized diagnostics for the over 6500 glioma patients diagnosed every year in Italy.

## 4. Materials and Methods

### 4.1. General

In the case of [^18^F]FET (IASOglio^®^), the manufacturing process has been re-developed to make it compatible with the AllinOne synthesizer. The synthesis, formulation, and transfer of the radiopharmaceutical [^18^F]FET (IASOglio^®^) bulk solution to the dispensing cell for injection, using nitrogen as propulsion gas, are performed using the Trasis AllinOne automated synthesizer, disposable cassette, and commercial reagent kit (Trasis, Rue Gilles Magnée 90, 4430 Ans, Belgium).

The synthesis duration of [^18^F]FET (IASOglio^®^) is about 55 min, with a general decay-corrected yield between 73 and 79%

The drug substance [^18^F]FET is manufactured by radiolabeling of the precursor O-(2-tosyloxyethyl)-N-trityl-L-tyrosine-tert-butyl ester (TET) in two synthetic steps using an automated system. The process includes eight production steps, described below. The crude active substance [^18^F]FET is purified using semi-preparative HPLC. The product collected following this purification is directly transferred to class A through a 0.22 µm filtration membrane. The substance obtained corresponds to the pharmaceutical substance [^18^F]FET.

The synthesis is described in Figure 1 and details of the production steps are provided hereafter:(1)Irradiation of the target consisting of oxygen-18-enriched water and transfer to the synthesis module.(2)Recovery of [^18^F]-fluorine and evaporation of the eluent: the [^18^F]-fluoride is adsorbed on an anion exchange resin, eluted from the resin, and then dried up.(3)Radiolabeling of the precursor: the precursor of the drug substance TET (1) reacts by nucleophilic substitution with the dried [^18^F]-fluoride to give the protected [^18^F]FET (2).(4)Deprotection of the labeled precursor: this intermediate (2) is hydrolyzed in acidic conditions to remove the protection groups and give [^18^F]FET.(5)HPLC purification: the crude [^18^F]FET (3) is then purified on a semi-preparative column.(6)Formulation and prefiltration in dispensing cell: a formulation buffer is added to the collected peak of [^18^F]-FET to stabilize the mother solution.(7)Calibration of the radiopharmaceutical via dilution and dispensing after sterilizing filtration.

The testing standards for the quality control of [^18^F]FET (IASOglio^®^) 2 GBq/mL solution for injection, manufactured on the AllinOne module, are aligned with or equivalent to the Ph. Eur. Monograph 2466 of Fluoroethyl-L-Tyrosine (18F) injection. Table 4 resumes the specifications of IASOglio^®^.

### 4.2. Technology Transfer Methodology

The first step towards the commercial production of a radiopharmaceutical on a PET production site is the technology transfer of the manufacturing process and quality control methods according to the current MA dossier and the current edition of European Pharmacopeia.

The technology transfer of [^18^F]FET (IASOglio^®^) at theCNR-IFC Pisa PET production site refers to the transfer of documentation, manufacturing processes, and analytical methods and can be summarized as follows:Gap analysis: at this stage, a thorough evaluation of equipment, documentation, and raw materials is carried out to plan or implement any necessary adjustments.Analytical Method Validation (AMV) and personnel training: validation of the whole package of analytical methods used for the quality control analysis of [^18^F]FET (IASOglio^®^) (i.e., HPLC, TLC, GC, Endotoxins, etc.) with theoretical and practical staff training.Validation/Transfer batch production and stability study: production and quality control analysis of three consecutive [^18^F]FET (IASOglio^®^) batches.Regulatory filing.Clinical supply.

Figure 2 shows the technology transfer steps for [^18^F]FET (IASOglio^®^).

#### 4.2.1. Gap Analysis

Gap analysis is the first step before deploying a new molecule on a production site. The gap analysis process leads to a list of actions to be performed until the validation of the process and subsequent start of commercial production.

Based on the gap analysis protocol, an evaluation of all equipment, raw materials, and documentation was carried out to ensure the proper installation of THE [^18^F]FET (IASOglio^®^) production process on the AllinOne module.

The objective of the gap analysis is to determine if a site is ready to welcome the manufacturing process for a new radiopharmaceutical (in this case, [^18^F]FET (IASOglio^®^)).

During this gap analysis, it was determined that all the equipment is already available on site, and no blocking actions have been identified.

In conclusion, at the end of the gap analysis, the manufacturing site was technically able to start the IASOglio^®^ production process on the AllinOne module.

#### 4.2.2. Analytical Method Validation and Staff Training

The AMV step refers to the entire process of ensuring that the facility is fully prepared to carry out the testing standards for the quality control of a new product.

As part of the AMV at the CNR-IFC Pisa PET production site, a risk assessment of the analytical method validation was performed to determine the requested validation activities needed to consider the facility capable of performing quality control testing of [^18^F]FET (IASOglio^®^).

After careful evaluation, the necessary analytical methods have been validated according to the conclusion of the risk assessment at the CNR-IFC Pisa PET production site using the analytical method validation protocols.

The analytical methods were validated as recommended by the International Conference of Harmonization (ICH) and are presented in Table 5 with an indication of the tests that must be performed. 

Staff training involves theoretical and practical training provided by the CMC team of Curium. Theoretical training consists of a presentation introducing the usefulness of [^18^F]FET (IASOglio^®^), as well as the theoretical aspects related to the synthesis and quality control of [^18^F]FET (IASOglio^®^). Practical training consists of the performance evaluation of [^18^F]FET (IASOglio^®^) batches, where the trainer will highlight the critical points of synthesis and quality control, giving an overview of the common cause of batch or analysis failures.

#### 4.2.3. Validation/Transfer Batch Production

[^18^F]FET (IASOglio^®^) analytical method and manufacturing process validation should provide documented evidence that the process delivers a final product complying with predetermined specifications in a reliable and reproducible manner by adhering to the process and maintaining the process parameters within the specified and examined range.

Qualification and validation activities are designed and executed according to previously approved protocols. Most notably, the protocols define objectives, methodologies, tests to be performed, limits, and acceptance criteria.

A final report specifically describes and analyzes the results obtained, conclusions generated, and consistency with the intended objectives.

Process validation is initiated exclusively after the successful completion of the following activities:Qualification of the premises;Qualification of the critical production equipment;Qualification of critical quality control equipment;Validation of analytical methods;Validation of cleaning procedures;Validation of washing of transfer lines;Qualification of information technology systems.

The process validation activities, subject to approval of the relevant protocols, are structured as follows:Execution of the bioburden test in triplicate;Execution of the media fill of the dilution and bottle filling system in triplicate;Repetition of the production process for three consecutive batches of finished product (also useful for documenting all the controlled parameters: radioactive dosage and calibration, quality controls, sterility and apyrogenicity of the finished product, and physical and microbiological environmental controls);Stability study of three batches, aimed to demonstrate compliance with the specifications of all the controlled parameters, within the validity period established for the product;Drafting and approval of the validation report.

During the validation/transfer of the [^18^F]FET (IASOglio^®^) radiopharmaceutical at the CNR-IFC Pisa PET production site, three consecutive batches were manufactured.

The study was performed on three batches of [^18^F]FET (IASOglio^®^) solution for injection, with the radiopharmaceutical manufactured at the CNR-IFC Pisa PET production site. The tests of chemical, physical, and microbial characteristics of [^18^F]FET (IASOglio^®^) solution for injection cover the shelf life of the product.

#### 4.2.4. Regulatory Filing

After the completion of process validation and transfer of analytical methods, an authorization request was submitted to the AIFA. After detailed verification of documentation, the AIFA issued a production authorization certificate, dated 4 April 2024 for the CNR-IFC Pisa PET production site, which is the first production authorization for [^18^F]FET (IASOglio^®^) issued in Italy.

In January 2025, the CNR-IFC Pisa PET production site was successfully added to the [^18^F]FET (IASOglio^®^) MA dossier, and an MRP is currently ongoing to start commercial production.

#### 4.2.5. Clinical Supply

The authorization process has been completed in Italy; however, large-scale distribution cannot begin until the French dossier is officially recognized by the Italian regulatory authorities. When the final step is completed, the radiopharmaceutical will be available in the national territory for large-scale distribution, ensuring that all citizens receive fair healthcare and improving the clinical accuracy of imaging tools for the diagnosis and follow-up of oncological patients.

In addition, the national availability of the radiopharmaceutical will facilitate the start of new clinical trials in combination with innovative therapeutic approaches, contributing to the continuous improvement of patient care in Italy.

## 5. Conclusions

Gliomas, especially the subtype glioblastoma multiforme, are among the most challenging and aggressive malignancies, despite representing only 3% of all malignant neoplasms. Their aggressive nature means that gliomas are responsible for a disproportionately high number of years of life lost compared to other malignancies.

The added value of PET with [^18^F]FET in the diagnosis and follow up of gliomas has been indicated by specialist guidelines; however, currently, in Italy, the availability of [^18^F]FET is limited, and the GMP compliant drug is only available if imported from abroad.

The successful regulatory processes and the release of the GMP certificate for [^18^F]FET (IASOglio^®^) mark a critical achievement in Italy’s healthcare system. The ability to manufacture and distribute [^18^F]FET on a national scale will ensure that all patients across the country, from the northern regions to the south of the country, will have access to this essential radiopharmaceutical. This development promises to improve the healthcare experience for glioma patients, offering them the hope of a more effective and personalized approach to their treatment and marking a significant leap forward in the fight against one of the most devastating forms of cancer.

## Figures and Tables

**Figure 1 pharmaceuticals-18-00769-f001:**
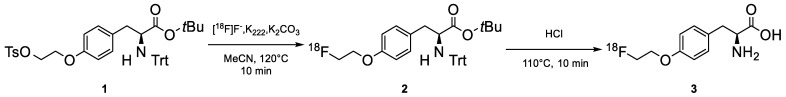
Reaction scheme and conditions for the preparation of the drug substance [18F]FET according to the Marketing Authorization production process.

**Figure 2 pharmaceuticals-18-00769-f002:**
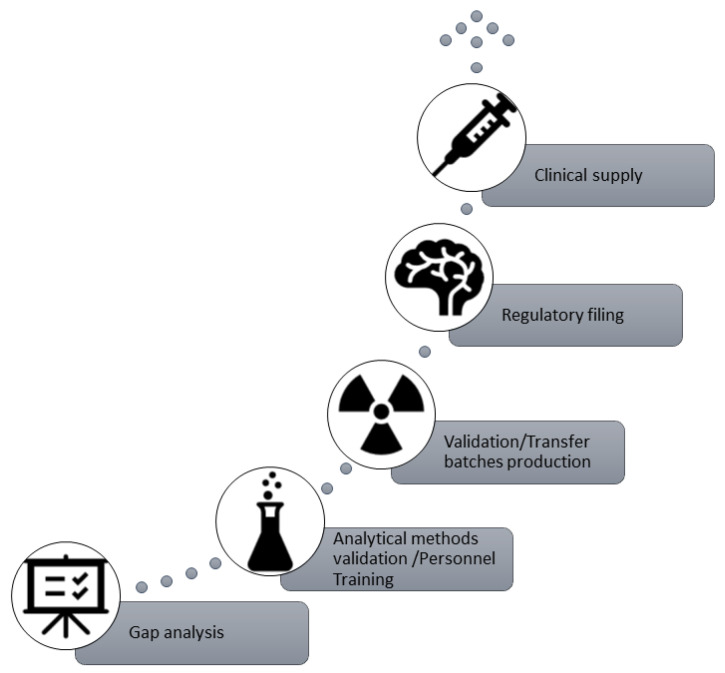
Technology transfer steps for [18F]FET (IASOglio^®^).

**Table 1 pharmaceuticals-18-00769-t001:** Quality control results for the three validation batches of IASOglio^®^.

Tests	Specifications	Results		
		Batch No. 1	Batch No. 2	Batch No. 3
Appearance	Clear and colorless to slightly yellow solution	Compliant	Compliant	Compliant
Identifications				
Test A: Energy of the γ photons	0.511 MeV and a peak sum of 1.022 MeV may be observed	0.511 MeV	0.511 MeV	0.511 MeV
Test B: Half-life	105–115 min	106 min	109 min	110 min
Test C: Radiochemical identification	Rt test solution similar to Rt reference solution (Rt standard ± 10%)	RT standard + 3%	RT standard + 3%	RT standard + 3%
Tests				
pH	4.5–8.5	6.5	5.3	5.8
Radionuclidic purity				
Energy of the γ photons	≥99.9%	100.0%	100.0%	100.0%
Radiochemical purity				
[18F]-FET	≥95%	≥95%	≥95%	≥95%
[18F]-Fluoride	≤5%	ND	ND	ND
Chemical purity				
Fluoroethyl-L-tyrosine	≤0.01 mg/mL	<0.01 mg/mL	<0.01 mg/mL	<0.01 mg/mL
Any other impurities	≤0.01 mg/mL	<0.01 mg/mL	<0.01 mg/mL	<0.01 mg/mL
Total of Fluoroethyl-L-tyrosine and impurities	≤0.05 mg/mL	<0.01 mg/mL	<0.01 mg/mL	<0.01 mg/mL
Kryptofix	≤0.22 mg/mL	<0.22 mg/mL	<0.22 mg/mL	<0.22 mg/mL
Residual solvent				
Acetonitrile	≤0.41 mg/mL	<0.41 mg/mL	ND	<0.41 mg/mL
Ethanol	≤79 mg/mL (10% (*v*/*v*))	≤79 mg/mL	≤79 mg/mL	≤79 mg/mL
Microbiological test				
Bacterial endotoxins	<17.5 IU/mL	<17.5 IU/mL	<17.5 IU/mL	<17.5 IU/mL
Radioactivity				
Radioactive concentration (at calibration)	2000 MBq/mL ± 10%	1981 MBq/mL	2036 MBq/mL	1977 MBq/mL
Test performed on decayed samples				
Long lasting radionuclidic impurities	Total radioactivity due to radionuclidic impurities ≤ 0.1%	<0.1%	<0.1%	<0.1%
Sterility	Sterile	Sterile	Sterile	Sterile

**Table 2 pharmaceuticals-18-00769-t002:** Stability results for three batches of IASOglio^®^.

Tests	Specifications	Results					
		Batch N. 1		Batch N. 2		Batch N. 3	
		At release	At expiry 40 °C ± 2 °C	At release	At expiry 40 °C ± 2 °C	At release	At expiry 40 °C ± 2 °C
Appearance	Clear and colorless to slightly yellow solution	Compliant	Compliant	Compliant	Compliant	Compliant	Compliant
Identifications							
Test A: Energy of the γ photons	0.511 MeV and a peak sum of 1.022 MeV may be observed	0.511 MeV	NA	0.511 MeV	NA	0.511 MeV	NA
Test B: Half life	105–115 min	106 min	NA	109 min	NA	110 min	NA
Test C: Radiochemical identification	Rt test solution similar to Rt reference solution (Rt standard ± 10%)	RT standard + 3%	RT standard + 1%	RT standard + 3%	RT standard + 2%	RT standard + 3%	RT standard + 3%
Tests							
pH	4.5–8.5	6.5	5.4	5.3	4.9	5.1	5.7
Radionuclidic purity							
Energy of the γ photons	≥99.9%	100.0%	NA	100.0%	NA	100.0%	NA
Radiochemical purity							
[^18^F]-FET	≥95%	100%	97%	97%	99%	100%	100%
[^18^F]-Fluoride	≤5%	ND	ND	ND	ND	ND	ND
Chemical purity							
Fluoroethyl-L-tyrosine	≤0.01 mg/mL	<0.01 mg/mL	<0.01 mg/mL	<0.01 mg/mL	<0.01 mg/mL	<0.01 mg/mL	<0.01 mg/mL
Any other impurities	≤0.01 mg/mL	<disregard limit	<disregard limit	<disregard limit	<disregard limit	<disregard limit	<disregard limit
Total of Fluoroethyl-L-tyrosine and impurities	≤0.05 mg/mL	<0.01 mg/mL	<0.01 mg/mL	<0.01 mg/mL	<0.01 mg/mL	<0.01 mg/mL	<0.01 mg/mL
Kryptofix	≤0.22 mg/mL	<0.22 mg/mL	NA	<0.22 mg/mL	NA	<0.22 mg/mL	NA
Residual solvent							
Acetonitrile	≤0.41 mg/mL	<0.41 mg/mL	NA	<0.41 mg/mL	NA	<0.41 mg/mL	
Ethanol	≤79 mg/mL (10% (*v*/*v*))	2% (*v*/*v*)	NA	1% (*v*/*v*)	NA	1% (*v*/*v*)	
Microbiological test							
Bacterial endotoxins	<17.5 IU/mL	<0.5 EU/mL	NA	<0.5 EU/mL	NA	<0.5 EU/mL	
Radioactivity							
Radioactive concentration (at calibration)	2000 MBq/mL ± 10%	1981 MBq/mL	NA	2036 MBq/mL	NA	2012 MBq/mL	
Test performed on decayed samples							
Long lasting radionuclidic impurities	Total radioactivity due to radionuclidic impurities ≤ 0.1%	<0.1%	NA	<0.1%	NA	<0.1%	
Sterility	Sterile	Sterile	NA	Sterile	NA	Sterile	

**Table 3 pharmaceuticals-18-00769-t003:** Results for the three bioburden batches of IASOglio^®^.

Test	Specifications	Results		
		Batch BB No. 1	Batch No. BB 2	Batch No. BB 3
Bioburden	Bacteria ≤ 1 CFU/10 mL	<1 CFU/10 mL	<1 CFU/10 mL	<1 CFU/10 mL
	Yeasts and molds ≤ 1 CFU/10 mL	<1 CFU/10 mL	<1 CFU/10 mL	<1 CFU/10 mL

**Table 4 pharmaceuticals-18-00769-t004:** Specifications of IASOglio^®^.

Tests	Methods	Specifications
Character		
Appearance	Visual inspection	Clear, colorless, or slightly yellow solution
Identification		
Test A: Energy of the γ photons	Gamma ray spectrometry (Ph. Eur 2.2.66)	0.511 MeV and a peak sum of 1.022 MeV may be observed
Test B: Half life	Gamma ray spectrometry or ionization chamber (Ph. Eur 2.2.66)	105–115 min
Test C: Radiochemical identification	Liquid chromatography (HPLC) (Ph. Eur 2.2.29)	Rt test solution similar to Rt reference solution (Rt standard ± 10%)
Tests		
pH	pH strip (Ph. Eur. 1178900) or pH-meter	4.5–8.5
Chemical purity		
Kryptofix	Thin-layer chromatography (TLC) (Ph. Eur. 2.2.27)	≤0.22 mg/mL
Fluoroethyl-L-tyrosine Any other impurities Total impurities	Liquid chromatography (HPLC) (Ph. Eur 2.2.29)	≤0.01 mg/mL
		≤0.01 mg/mL
		≤0.05 mg/mL
Radionuclidic purity		
Energy of the γ photons	Gamma ray spectrometry (Ph. Eur 2.2.66)	≥99.9%
Radiochemical purity		
[^18^F]-FET	Liquid chromatography (HPLC) (Ph. Eur 2.2.29)	≥95%
[^18^F]-Fluoride	Thin-layer chromatography (TLC) (Ph. Eur 2.2.27)	≤5%
Residual solvents		
Acetonitrile	Gas chromatography (GC) (Ph. Eur 2.2.28)	≤0.41 mg/mL
Ethanol content		≤79 mg/mL (10% (*v*/*v*))
Microbiological test		
Bacterial endotoxin	Chromogenic kinetic method or turbidimetric method (Ph. Eur 2.6.14)	<17.5 EU/mL
Radioactivity		
Radioactive concentration (at calibration)	γ-spectrometry or ionization chamber counter (Ph. Eur 2.2.66)	90–110% of the announced radioactivity
Test performed on decayed samples		
Long lasting radionuclidic impurity	Gamma ray spectrometry (Ph. Eur 2.2.66)	Total radioactivity due to radionuclidic impurities ≤ 0.1%
Sterility	Direct inoculation (Ph. Eur. 2.6.1 and 0125)	Sterile

**Table 5 pharmaceuticals-18-00769-t005:** ICH Q2 (R1) validation of radiopharmaceuticals.

TEST	INSTRUMENT						
	Ionization Chamber	Gamma Spectrometer		HPLC		TLC	GC
		Identity	Radionuclide Purity	Chemical Purity	Radiochemical Purity		
Accuracy	✔	✔	✔	✔	✔	✔	✔
Precision (Repeatability)	✔	-	-	✔	✔	✔	✔
Intermediate Precision	-	-	-	✔	✔	✔	✔
Specificity	✔	✔	✔	✔	✔	✔	✔
Limit of Detection (LOD)	-	-	✔	✔	✔	✔	✔
Limit of Quantification (LOQ)	-	-	-	✔	✔	✔	✔
Linearity	✔	-	-	✔	✔	✔	✔
Range	✔	-	-	-	✔	✔	-
Robustness	-	-	-	✔	✔	✔	✔

## Data Availability

The data presented in this study are available in this paper.

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
