# Peer review of "Technology Transfer of O-(2-[18F] Fluoroethyl)-L-Tyrosine (IASOglio®) Radiopharmaceutical"

_pharmaceuticals, 2025, doi:10.3390/ph18060769_

Round 1

Reviewer 1 Report

Comments and Suggestions for Authors

The manuscript altogether seems to be an useful and timely contribution especially in Italy with the aim to improve national access to this advanced PET diagnostic technology.

Based on the novel data presented in the MS and the useful value of their findings including innovative approaches that can be beneficial for Italian glioblastoma patents I recommend to accept the manuscript for publication. However, some minor changes are needed and also some additional data would be required before accepting the manuscript in its final version.

  1. The affiliation section looks funny, they need to revise it. There are even missing locations.
  2. The results section of the MS seems to be very  superficial. They need to present more details about their findings adding figures, tables corresponding to their results.

Author Response

Comments 1: The affiliation section looks funny, they need to revise it. There are even missing locations.

Response 1: We agree with this comment. Therefore, we have modified the affiliation with missing information (location and zip code). Page 1 lines 6-13.

Comments 2: The results section of the MS seems to be very superficial. They need to present more details about their findings adding figures, tables corresponding to their results.

Response 2: We agree with this comment. Therefore, we have added some tables with the detailed results. Table 1 Page 4 line 161, Table 2 Page 5 line 167, Table 3 Page 6 line 173.

Reviewer 2 Report

Comments and Suggestions for Authors

This paper describes the technological transfer of the radiopharmaceutical [18F]FET (IASOglio®) to improve its diagnostic effect. However, the manuscript describes technological transformation, which could not be considered as a type of research article. In this point, I would suggest the reconsideration of the manuscript after addressing the following comments:

1 No figure or table could be found in Results part. Please add related figures or tables to support your conclusion.

2 How to synthesize compound 1, 2, and 3? Please label the reaction conditions in the caption part of Figure 1

3 Now that this is a radiopharmaceutical for gliomas, please show detailed results of [18F]FET (IASOglio®) applied in imaging gliomas tumors in patients.

Author Response

Comments 1: No figure or table could be found in Results part. Please add related figures or tables to support your conclusion.

Response 1: We agree with this comment. Therefore, we have added some tables with detailed results. Table 1 Page 4 line 161, Table 2 Page 5 line 167, Table 3 Page 6 line 173.

Comments 2 How to synthesize compound 1, 2, and 3? Please label the reaction conditions in the caption part of Figure 1

Response 2: We have, accordingly, modified the figure 1 with the addition of the reaction conditions. Figure 1 Page 9 line 277

Comments 3 Now that this is a radiopharmaceutical for gliomas, please show detailed results of [18F]FET (IASOglio®) applied in imaging gliomas tumors in patients.

Response 3: We agree with this comment, so we have added a paragraph concerning the results of [18F]FET applied in imaging gliomas tumors in patients. Page 2 Lines 57-64.

The information relating to the results of [18F]FET applied in imaging gliomas tumors in patients is reported in the paper “Response Assessment in Neuro-Oncology working group and European Association for Neuro-Oncology recommendations for the clinical use of PET imaging in glioma” (Albert et al.). The reference has been included in the references section. Pag 16 Lines 473-476

Reviewer 3 Report

Comments and Suggestions for Authors

Dear Authors,

Thank you for the opportunity to review your study. This research addresses an important and interesting subject: improving the early diagnosis of gliomas. This is a crucial medical topic with significant implications for the general population. However, the paper requires major revision. Although the results and study design are well executed, the text is written by specialists for other specialists in the same field. Your valuable findings need to be communicated clearly to reach a broader audience, including non-specialists. My personal opinion is that the current presentation makes it challenging to convey your fantastic study to those outside your research area. It is advisable to focus on this aspect of your work.

I will provide detailed comments below.

Thank you again for your excellent research.

Regards,

Reviewer.

Comments/Suggestions.

Abstract.
Abbreviations should be defined in the text only if they are used more than once. Some abbreviations were defined once in the abstract and not used again, such as PET. Others, like [18F]FET or IASOglio®, were not defined at all. Please review all abbreviations in the abstract and main text of the paper. Make sure to include them in the abbreviation list at the end of the paper.

Line 44.
Add the abbreviation "[18F]FET" to the abbreviations list at the end of your manuscript.

Line 50.
Add the abbreviation "EANM" to the abbreviations list at the end of your manuscript.

Line 51.
Add the abbreviation "SNMMI" to the abbreviations list at the end of your manuscript.

Line 53.
Add the abbreviation "PET-RANO" to the abbreviations list at the end of your manuscript.

Line 61.
Add the abbreviation "GMP" to the abbreviations list at the end of your manuscript.

Line 65.
The text discusses a marketing authorization that is referenced in citation [8] of the references list. This citation leads to a webpage link from which a PDF file in French can be accessed. This poses a challenge for readers who are not familiar with the French language. It would be beneficial if the authors included a summary of the technical details relevant to this study within the manuscript.

Line 80.
Comparing the synthesis duration of 60 minutes for producing [18F]FET to the half-life of Fluorine-18, which is 109.771 minutes, shows that the synthesis process is considerably long relative to the isotope’s short half-life. By detailing this comparison, readers can better understand the challenges in optimizing production to make the most of Fluorine-18's brief availability while maintaining efficiency and effectiveness in the synthesis process.

Line 95.
Add the abbreviation "CMC" to the abbreviations list at the end of your manuscript.

Line 101.
The authors discuss the shelf life of the product at the activity concentration level as outlined in the marketing authorization dossier referenced in citation [14]. However, readers of this manuscript do not have access to this document, and a quick internet search does not retrieve it online. It would be beneficial for the authors to provide this document on a free online server (such as arXiv) or include it as supplementary material for the paper. Alternatively, key technical details from the document should be incorporated into the main text for greater clarity.

Line 102.
The abbreviation "CMC" is defined in line 95; please refrain from repeating it unnecessarily.

Line 108.
The abbreviation "CNR-IFC" is explained in the abstract, but not when it first appears in the main text. Please add the full form there as well, following the journal’s guidelines.  Also, add the abbreviation "CNR-IFC" to the abbreviations list at the end of your manuscript. Here is citations from preparation of manuscript section for this journal: "Acronyms/Abbreviations/Initialisms should be defined the first time they appear in each of three sections: the abstract; the main text; the first figure or table. When defined for the first time, the acronym/abbreviation/initialism should be added in parentheses after the written-out form."

Lines 121, 122.
For the study's purposes, it would be helpful to include details about the methods used to perform the decay corrections. By providing this information, the study can offer a comprehensive understanding of the processes involved, enhancing the validity and replicability of the results.

Lines 127 to 132.
The paragraph claims that the radiochemical purity of the product consistently exceeded 95%. However, the statement lacks supporting details such as the margin of uncertainty, testing methods, or verification processes utilized. Moreover, as previously noted, readers do not have access to the marketing authorization dossier mentioned in citation [14]. Consequently, more comprehensive details supporting the claim in this paragraph are necessary.

Lines 133 to 142.
The section summarizes the results of various verifications and confirms compliance with the MA dossier and European Pharmacopeia. However, it would strengthen the manuscript to briefly describe how the results were verified—for example, by indicating the methods, instruments, or protocols used. This would improve the transparency and reproducibility of the reported quality assurance process.

Lines 179 to 183.
The documentary evidence being discussed is not specified. Additionally, the specifications of the final products that were tested are unclear. It is also uncertain which process parameters were maintained within the examined range and what those range specifications entail.

Lines 184 to 200.
This section describes the successful transfer of technology. In the transfer process, there is a "receiver" and a "sender." The receiver of the technology is the CNR-IFC Pisa PET production site. However, it is unclear who the "sender" of this technology is.

Lines 209,210.
It is unclear whether the general yield has been adjusted for decay.

Line 241.
It would be beneficial for readers if authors include a reference to 'Ph. Eur. Monograph 2466' which discuss the Fluoroethyl-L-Tyrosine (18F) injection.

Section 4.2.1. Gap analysis
It is not clear what the outcome of the gap analysis process was. Were there any findings that needed to be addressed? Did the analysis conclude that the production site is ready for this product? Please consider adding more details regarding the findings and outcomes of the gap analysis.

Section 4.2.2. Analytical methods validation and staff training
The analytical method validation protocols are not clearly described in this section. Kindly provide additional technical details to support the methods used and ensure clarity.

Section 4.2.3. Validation/Transfer batches production
The original text does not include technical details on the validation process. Specifically, it lacks information on the documentary evidence of the transfer process and how readers of this manuscript can access this evidence. Furthermore, it does not present the test results for the chemical, physical, and microbial characteristics of [18F]FET (IASOglio®).

Section 4.2.5. Clinical supply
The statement claims: "Once the authorization process is completed, the radiopharmaceutical will be available for large-scale distribution across the national territory, ensuring equitable healthcare access, enhancing the clinical accuracy of imaging tools for diagnosing and monitoring oncological patients." However, this contradicts the previous section, "4.2.4. Regulatory Filing," which mentions that the authorization process was completed in January 2025. There is ambiguity regarding the current (April 2025) availability of this product for clinical use. For instance, if a clinic requests the product from a new production site that has received technology transfer, it is unclear whether this product can be purchased.

Comments on the Quality of English Language

The authors received feedback in my reviewer notes regarding the quality of the English language used in their work. 

Author Response

Comments 1:  Abstract. Abbreviations should be defined in the text only if they are used more than once. Some abbreviations were defined once in the abstract and not used again, such as PET. Others, like [18F]FET or IASOglio®, were not defined at all. Please review all abbreviations in the abstract and main text of the paper. Make sure to include them in the abbreviation list at the end of the paper.

We agree with this comment, all the suggestion of the reviewer are accepted as reported below:

Line 44.
Add the abbreviation "[18F]FET" to the abbreviations list at the end of your manuscript.

"[18F]FET" has been added to the abbreviations list Pag 14 Line 445

Line 50.

Add the abbreviation "EANM" to the abbreviations list at the end of your manuscript.

"EANM" has been added to the abbreviations list Pag 14 Line 445

Line 51.
Add the abbreviation "SNMMI" to the abbreviations list at the end of your manuscript.

"SNMMI" " has been added to the abbreviations list Pag 14 Line 445

Line 53.
Add the abbreviation "PET-RANO" to the abbreviations list at the end of your manuscript.

"PET-RANO" " has been added to the abbreviations list Pag 14 Line 445

Line 61.
Add the abbreviation "GMP" to the abbreviations list at the end of your manuscript.

"GMP" " has been added to the abbreviations list Pag 14 Line 445

Line 65.
The text discusses a marketing authorization that is referenced in citation [8] of the references list. This citation leads to a webpage link from which a PDF file in French can be accessed. This poses a challenge for readers who are not familiar with the French language. It would be beneficial if the authors included a summary of the technical details relevant to this study within the manuscript.

We have, accordingly, translate the document in English and it is added in the section of supplementary materials. In the references section the documents is listed with [9]

Line 80.
Comparing the synthesis duration of 60 minutes for producing [18F]FET to the half-life of Fluorine-18, which is 109.771 minutes, shows that the synthesis process is considerably long relative to the isotope’s short half-life. By detailing this comparison, readers can better understand the challenges in optimizing production to make the most of Fluorine-18's brief availability while maintaining efficiency and effectiveness in the synthesis process.

We agree with this comment, so we have added a sentence in the introduction. Pag 3 Line 101-106.

Also, we have added a sentence in the Discussion section to infantize the importance of optimizing the production time maintaining efficiency and effectiveness in the synthesis process. Pag 7 Line 215-218

Line 95.
Add the abbreviation "CMC" to the abbreviations list at the end of your manuscript.

"GMP" " has been added to the abbreviations list Pag 14 Line 445

Line 101.
The authors discuss the shelf life of the product at the activity concentration level as outlined in the marketing authorization dossier referenced in citation [14]. However, readers of this manuscript do not have access to this document, and a quick internet search does not retrieve it online. It would be beneficial for the authors to provide this document on a free online server (such as arXiv) or include it as supplementary material for the paper. Alternatively, key technical details from the document should be incorporated into the main text for greater clarity.

The information contained in the marketing authorization dossier cannot be disclosed, as it is considered confidential. We have added in the supplementary material the “Resume-des-Caracteristiques-Produit-IASOglio” translated in English and listed in the references section with [9] number.

Line 102.
The abbreviation "CMC" is defined in line 95; please refrain from repeating it unnecessarily.

The abbreviation CMC has been used. Pag 3 Line 114

Line 108.
The abbreviation "CNR-IFC" is explained in the abstract, but not when it first appears in the main text. Please add the full form there as well, following the journal’s guidelines.  Also, add the abbreviation "CNR-IFC" to the abbreviations list at the end of your manuscript. Here is citations from preparation of manuscript section for this journal: "Acronyms/Abbreviations/Initialisms should be defined the first time they appear in each of three sections: the abstract; the main text; the first figure or table. When defined for the first time, the acronym/abbreviation/initialism should be added in parentheses after the written-out form."

The abbreviation "CNR-IFC" has been added in the main text. Pag 3 Lines 120- 121

"CNR-IFC" " has been added to the abbreviations list Pag 14 Line 445

Lines 121, 122.
For the study's purposes, it would be helpful to include details about the methods used to perform the decay corrections. By providing this information, the study can offer a comprehensive understanding of the processes involved, enhancing the validity and replicability of the results.

We agree with this comment, the decay-corrected yield provides standardized information about the synthesis regardless of the decay time of the radioisotope, so we have added a sentence o better explain this concept. We also inserted formula the decay calculation. Pag 3 Lines 134-142

Lines 127 to 132.
The paragraph claims that the radiochemical purity of the product consistently exceeded 95%. However, the statement lacks supporting details such as the margin of uncertainty, testing methods, or verification processes utilized. Moreover, as previously noted, readers do not have access to the marketing authorization dossier mentioned in citation [14]. Consequently, more comprehensive details supporting the claim in this paragraph are necessary.

We agree with this comment. Therefore, we have added some tables with the detailed results. Table 1 Page 4 line 161, Table 2 Page 5 line 167, Table 3 Page 6 line 173 and we also added table 4 relating to the testing methods and to the specifications. Table 4 Page 9 line 284

Lines 133 to 142.
The section summarizes the results of various verifications and confirms compliance with the MA dossier and European Pharmacopeia. However, it would strengthen the manuscript to briefly describe how the results were verified—for example, by indicating the methods, instruments, or protocols used. This would improve the transparency and reproducibility of the reported quality assurance process.

We agree with this comment. Therefore, we have added some tables with the detailed results. Table 1 Page 4 line 161, Table 2 Page 5 line 167, Table 3 Page 6 line 173 and we also added table 4 relating to the testing methods and to the specifications. Table 4 Page 9 line 284.

Lines 179 to 183.
The documentary evidence being discussed is not specified. Additionally, the specifications of the final products that were tested are unclear. It is also uncertain which process parameters were maintained within the examined range and what those range specifications entail.

We agree with this comment, so we have added table 4 relating to the testing methods and to the specifications of the final product. Table 4 Page 9 line 284.

Lines 184 to 200.
This section describes the successful transfer of technology. In the transfer process, there is a "receiver" and a "sender." The receiver of the technology is the CNR-IFC Pisa PET production site. However, it is unclear who the "sender" of this technology is.

We agree with this comment; the sender is Curium Pharma. We added a sentence to specify that the transfer of technology happened in accordance with the indication provided by Curium Pharma. Page 7 line 229-230.

Lines 209,210.
It is unclear whether the general yield has been adjusted for decay.

We agree with this comment; the yield has been adjusted for decay Page 8 line 250

Line 241.
It would be beneficial for readers if authors include a reference to 'Ph. Eur. Monograph 2466' which discuss the Fluoroethyl-L-Tyrosine (18F) injection.

We agree with this comment; a reference to 'Ph. Eur. Monograph has been added Page 10 line 291.

Section 4.2.1. Gap analysis
It is not clear what the outcome of the gap analysis process was. Were there any findings that needed to be addressed? Did the analysis conclude that the production site is ready for this product? Please consider adding more details regarding the findings and outcomes of the gap analysis.

We agree with this comment; a sentence has been added to resume the Gap analysis results Page 11 lines 317-320

Section 4.2.2. Analytical methods validation and staff training
The analytical method validation protocols are not clearly described in this section. Kindly provide additional technical details to support the methods used and ensure clarity.

We agree with this comment; The analytical methods are validated as recommended by International Conference of Harmonization. A table with a clarification of the test applied in analytical methods validation has been added. Table 5 Page 12 line 335

Section 4.2.3. Validation/Transfer batches production
The original text does not include technical details on the validation process. Specifically, it lacks information on the documentary evidence of the transfer process and how readers of this manuscript can access this evidence. Furthermore, it does not present the test results for the chemical, physical, and microbial characteristics of [18F]FET (IASOglio®).

We agree with this comment, a description of the technical details on the validation process has been added. Page 12-13 lines 351-377.

Section 4.2.5. Clinical supply
The statement claims: "Once the authorization process is completed, the radiopharmaceutical will be available for large-scale distribution across the national territory, ensuring equitable healthcare access, enhancing the clinical accuracy of imaging tools for diagnosing and monitoring oncological patients." However, this contradicts the previous section, "4.2.4. Regulatory Filing," which mentions that the authorization process was completed in January 2025. There is ambiguity regarding the current (April 2025) availability of this product for clinical use. For instance, if a clinic requests the product from a new production site that has received technology transfer, it is unclear whether this product can be purchased.

Thank you for your comment. We acknowledge the opportunity to clarify. The authorization process has been completed in Italy, however, large-scale distribution cannot begin until the French dossier is officially recognized by the Italian regulatory authorities.

Therefore, although the radiopharmaceutical is authorized, nationwide availability is pending the final regulatory alignment, which is currently in progress. We have added a sentence to better explain the situation. Page 13 lines 394-396.

Round 2

Reviewer 2 Report

Comments and Suggestions for Authors

Accept in present form

Author Response

Dear Reviewer,

Thank you very much for taking the time to review this manuscript.

Reviewer 3 Report

Comments and Suggestions for Authors

Dear Authors,

Thank you for your thorough revisions and for addressing all scientific and methodological concerns. From a reviewer’s perspective, the manuscript is now sound and requires only a minor revision before final acceptance.

A few remaining editorial items should be handled by the authors or the journal’s production team:

Table formatting consistency: Apply a uniform style across all tables (e.g., consistent header formatting and line rules).

Sentence spacing: Add a space after each period before starting the next sentence (e.g., line 76).

Table 1 pagination: Table 1 is split across two pages; if possible, please place it on a single page for readability.

Table 2 layout: Align fonts and columns to match the other tables for a coherent presentation.

Table 4 asterisks: Some entries in Table 4 include asterisks, but no footnote explains their meaning—please add the appropriate legend.

Math styling for equations: Ensure that all equations are typeset in proper math mode for consistency and readability.

Once these minor editorial and formatting issues are resolved, I fully support publication of your work.

Best regards,

Reviewer

Author Response

Dear Reviewer,

Thank you very much for taking the time to review this manuscript. We have tried to respond to your suggestions. Following the responses to your suggestions

Comment 1: Table formatting consistency: Apply a uniform style across all tables (e.g., consistent header formatting and line rules).

Response 1: Thank you, I agree with your comment, all tables have been reviewed and the style was aligned

Comment 1: Sentence spacing: Add a space after each period before starting the next sentence (e.g., line 76).

Response 1: Thank you, the text has been formatted following the reviewer indication

Comment 1: Table 1 pagination: Table 1 is split across two pages; if possible, please place it on a single page for readability.

Response 1: Thank you, table 1 is now in a single page

Comment 1: Table 2 layout: Align fonts and columns to match the other tables for a coherent presentation.

Response 1: Thank you, all the tables have been uniformed

Comment 1: Table 4 asterisks: Some entries in Table 4 include asterisks, but no footnote explains their meaning—please add the appropriate legend.

Response 1: Thank you, all asterisks have been removed

Comment 1: Math styling for equations: Ensure that all equations are typeset in proper math mode for consistency and readability.

Response 1: Thank you, the equation has been reviewed